

# Comparative study of nasal cavity drug delivery efficiency with different nozzles in a 3D printed model

Shengjian Fang[1,*], Xiaoqing Rui[1,*], Yu Zhang[1], Zhangwei Yang[2] and Weihua Wang[1]

[1] Department of Otolaryngology-Head and Neck Surgery, Shanghai East Hospital, School of Medicine, Tongji University, Shanghai, China
[2] Department of Medical Imaging, Shanghai East Hospital, School of Medicine, Tongji University, Shanghai, China
[*] These authors contributed equally to this work.

Corresponding author
Weihua Wang,
whwangcn@tongji.edu.cn

## ABSTRACT

**Background**. Nasal sprays are widely used in treating nasal and sinus diseases; however, there are very few studies on the drug delivery efficiency of nasal sprays. In this study, the drug delivery efficiency of three different nasal spray devices was evaluated *in vitro* using a 3D printed cast model of nasal cavity.

**Methods**. Three nasal spray devices with different nozzles and angles of administration were used in the 3D model of the nasal cavity and paranasal sinuses. The spraying area (SA), maximal spraying distance (MSD), and spraying distribution scores on the nasal septum and lateral nasal wall were recorded.

**Results**. Different nasal spray devices have their own characteristics, including volume of each spray, SA, and plume angle. The SA of the three nozzles on the nasal septum increased with an increasing angle of administration. When the angle of administration was 50°, each nozzle reached the maximal SA. There was no statistically significant difference in MSD among the three nozzles at the three angles. The total scores for each nozzle using the three different spraying angles were as follows: nozzle A, 40° > 30° > 50°; nozzle B, 30° > 40° > 50°; and nozzle C, 30° > 40° > 50°. The total scores for different nozzles using the same angle were statistically significantly different and the scores for nozzle C were the highest. Nozzle C had the minimum plume angle. None of the three nozzles could effectively delivered drugs into the middle meatus at any angle in this model.

**Conclusions**. The design of the nozzle affects drug delivery efficiency of nasal spray devices. The ideal angle of administration is 50°. The nozzle with smaller plume angle has higher drug delivery efficiency. Current nasal spray devices can easily deliver drugs to most areas of the nasal cavity, such as the turbinate, nasal septum, olfactory fissure, and nasopharynx, but not the middle meatus. These findings are meaningful for nozzle selection and device improvements.

## INTRODUCTION

Chronic rhinosinusitis (CRS) and allergic rhinitis (AR) are common upper airway diseases. It has been estimated that CRS and AR affect 12.5% and 33% of the global population, respectively (*Dykewicz & Hamilos, 2010*; *Meltzer et al., 2004*). The advantage of airways is the accessibility to topical administration of treatments (*Varricchio et al., 2023*). In addition to CRS and AR, nasal administration is also an ideal way for the management of ailments such as nasal congestion and common cold symptoms (*Forbes et al., 2020*). The intranasal vaccines confer a longer duration and better cross protection than the conventional injectable vaccines (*Zaman, Chandrudu & Toth, 2013*). The nasal cavity is covered with a highly vascularized and relatively permeable mucosa. The presence of immunocompetent cells in the nasal mucosa makes it a potential target organ for the nasal treatment (*Forbes et al., 2020*). Moreover, the subtle relationship between the relatively large mucosal surface area and the small volume of nasal cavity also promotes sufficient interaction between drugs and the mucosa. This is why drugs delivered through the nasal cavity can be absorbed quickly and has a therapeutic effect (*Djupesland, 2013*). Intranasal administration is considered to be a more effective way to reduce side effects of systematic bioavailability (*Bleske et al., 1992*; *Djupesland, 2013*; *Ting, Gonda & Gipps, 1992*; *Varricchio et al., 2023*). Several delivery devices are available on the market, and doctors should choose the most appropriate device for the current disease in clinical practice (*Varricchio et al., 2023*). In addition, direct access to the brain through the olfactory area is another advantage of nasal administration (*Forbes et al., 2020*).

A variety of factors affect the optimization of sinonasal drug delivery, one of which is the pattern of intranasal drug deposition (*Djupesland, Messina & Mahmoud, 2020*; *Foo et al., 2007*; *Sosnowski et al., 2020*). Delivery differences may be more important for topically acting medications (*Obaidi et al., 2013*). Inspiratory airflow spreads the deposited drug more distally into the nasal cavity (*Sosnowski et al., 2020*). Many devices and formulations have been well established for nasal delivery (*Costantino et al., 2007*; *Djupesland, 2013*; *Forbes et al., 2020*). Because of the convenience and dose consistency, nasal sprays have become a routine method of intranasal administration. This process is usually achieved by delivering a conical spray of drug solutions or suspensions into the nasal cavities using a hand-actuated metered dose pump. The effective range of distribution of nasal sprays in the nasal cavity depends on the dynamic nasal anatomic structure during normal inspiration/expiration and the nasal cycle (*Baraniuk, 2008*; *Djupesland & Skretting, 2012*), the geometry of the aerosol cloud (*Sosnowski et al., 2020*), spray technique, nasal deposition and clearance, dosage form, drive mode, and nozzle design (*Bateman et al., 2002*; *Hallworth & Padfield, 1986*; *Hardy, Lee & Wilson, 1985*; *Harris et al., 1988*; *Newman, Moren & Clarke, 1988*; *Newman, Moren & Clarke, 1987*; *Suman, Laube & Dalby, 1999*). Drug spraying range is the key parameter used to evaluate drug delivery efficiency. Previous studies have shown that the main factors affecting spray droplet entry into the nasal cavity are spray velocity, angle, particle size, surface tension, and viscosity (*Cheng et al., 2001*; *Dong et al., 2020*; *Foo et al., 2007*; *Sosnowski et al., 2020*). To the best of our knowledge, most studies have semi-quantitatively evaluated nasal delivery efficiency by comparing the amount of marker

recovered within the nasal cavity to the total amount of marker emitted from the device for each actuation (*Foo et al., 2007*). The specific distribution of drugs in the nasal cavity is still not fully understood and the drug delivery efficiency of different devices was not quantitatively compared.

Nasal casts are effective tools to study nasal delivery efficiency, which may characterize intranasal drug deposition (*Djupesland, Messina & Mahmoud, 2020*). In this study, a 3D printed model of a healthy adult nasal cavity and paranasal sinus was created and three frequently used devices in China were tested. These devices have different nozzle designs, with their own characteristic spray plumes. We quantitatively compared the drug delivery efficiency of these nozzles by measuring the spraying area (SA) and the maximal spraying distance (MSD) in this model.

## MATERIALS & METHODS

### Study design

Three hand-actuated pump nasal spray devices frequently used in China were chosen in this study: nasal spray device A (Daphne Pharmaceutical Co., Ltd.), B (Shanghai Johnson & Johnson Pharmaceuticals, Ltd.), and C (Glaxo Wellcome S.A.). A colored spraying solution was prepared by mixing normal saline with a blue-eluting pigment (Methylene blue; Sigma-Aldrich, MO, USA) at a ratio of 5:1 to replace the corresponding contents in the spray devices. A 3D printed model was created based on the computed tomography (CT) data from a 37-year-old healthy female volunteer without nasal disorder. The simulation application of these spray devices was realized using this model. The SA and MSD of the three nozzles were recorded and compared. Written informed consent was obtained from this volunteer.

### Production of 3D printed nasal cast model

The healthy volunteer had no obvious anatomic variations in the nasal cavity and paranasal sinuses, and no history of sinonasal disease. To improve the model simulation, the thicknesses of the CT scan and reconstruction layer were set at 0.5 mm (Fig. 1). The image data in DICOM format acquired from CT scan was exported. The segmentation module was utilized to separate the external nose, nasal cavity, and paranasal sinuses from the skull. DICOM data of the nasal cavity and paranasal sinuses were divided into four cuboid components. For 3D reconstruction, the format of each component of data was then converted into STL files using Mimics software (version 20; Materialise NV, Leuven, Belgium). The 3D model was printed with a 3D printer (UPBox, Tiertime Technology Co., Beijing, China) using polylactic acid (in a diameter of 1.75 mm; Shenzhen Esun Industrial Co., Ltd., Shenzhen, China) with a printing duration of 12 h (Fig. 2A). The printer was configured with the following parameters: layer thickness of 0.2 mm, fill density of 100%, print head diameter of 0.4 mm, nozzle temperature of 210 °C, print speed of 30 mm/s, shrinkage compensation enabled, cooling fan activated, and housing thickness set to 0.8 mm. For convenience of observation, the nasal septum of the model was removed. Structures on the lateral nasal wall were clearly visible (Fig. 2B). The model faithfully reproduced the structures of the human nasal cavity and the paranasal sinus. In this nasal

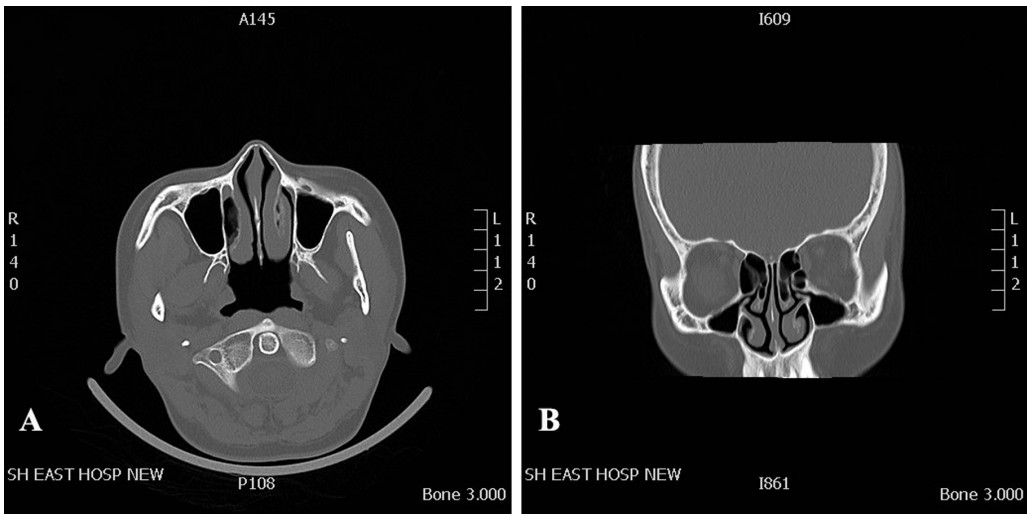

**Figure 1** **CT data that was used for 3D printed model.** CT scan of a 37-year-old female volunteer without nasal disorder. (A) Axial plane. (B) Coronal plane.

cast model, the angle between hard palate horizontal plane and the extension of the line from the highest point of the internal nasal valve was close to 60°.

## Spray parameters measurement

The nasal spray devices were actuated by an automatic actuator (model: L729, DC 2–5 V; Weibo Electrical, Changzhou, China) (Fig. 3). The displacement distance of the push arm is 3.0 cm, the maximum thrust 50 N, and the maximum stretching frequency 130 times/min. The velocity of automatic actuator is 0.2 m/s. Each nasal spray device was placed vertically upward and actuated three times to prime the device, followed by one test actuation. Spraying range measurements were made at a distance of four cm between the recording paper and the nozzle. The SA of each test on recording paper was scanned and imported into AutoCAD2018 software (Autodesk, San Rafael, CA, USA) and calculated automatically. Each nozzle was measured six times to calculate the mean value. The plume angle images were taken along the centerline of the device parallel to the recording paper. In all tests, images were captured at 60 frames/sec, and the maximum plume angle per test was measured and recorded.

## Simulation of nasal spray administration

Before intranasal administration, a piece of A4 size sulfuric acid transparent paper was placed in the sagittal plane to replace the nasal septum. Different angles with hard palate horizontal plane (30°, 40°, and 50°) were marked on the transparent paper. The 3D cast model was placed on a stable table. During each test, the angle of the nozzle was adjusted to be consistent with the line marked on the paper (Figs. 2C–2E). The nozzle direction was parallel to the sagittal plane. The insertion depth of the nozzle inside the nasal vestibule was set at 1.5 cm. The model was cleaned and dried after each test. Each device was tested nine times for each angle.

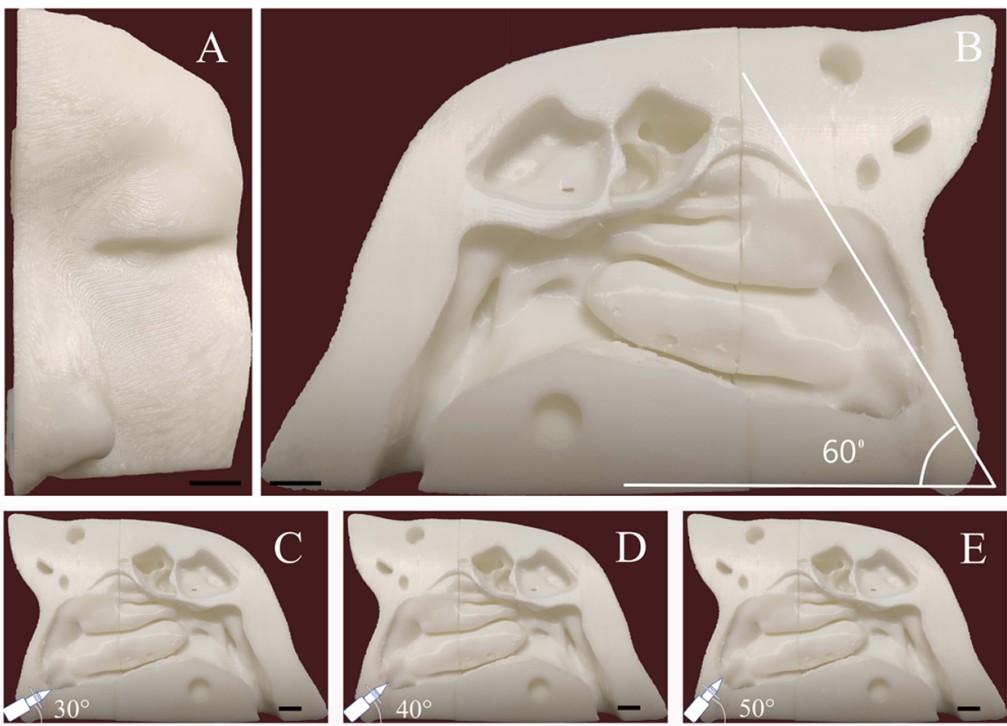

**Figure 2** **Three-dimensional printed model the adult nasal cavity and paranasal sinus.** (A) Anterior view. (B) Medial aspect view. The angle between the nasal base and the line connecting the highest point passing through the internal nasal valve was 60°. (C, D, and E) Schematic diagram of the administration angle. The insertion depth of each nozzle inside the nasal vestibule was 1.5 cm. Using the nasal base as the horizontal line, the nozzle angle was adjusted to 30°, 40°, or 50°. The scale bars represent 1 centimeter.

## Scoring of spray distribution on the lateral nasal wall

The distribution of nasal sprays on the lateral nasal wall was assessed using a scoring system. These regions on the lateral nasal wall include inferior turbinate, middle turbinate, inferior meatus, middle meatus, olfactory cleft, sphenoethmoidal recess, and nasopharynx. After each test, images of the staining distribution on the lateral nasal wall were captured using a digital camera under standardized photographic conditions. Scores were given based on the region reached by spray staining and the scoring criteria were as follows: anterior part of the inferior turbinate (score 1), posterior part of the inferior turbinate (score 2), middle turbinate (score 2), middle meatus (score 3), inferior meatus (score 1), olfactory cleft (score 2), sphenoethmoidal recess (score 2), and nasopharynx (score 2). Each device was tested nine times for each angle. As long as there was obvious staining in the region, the corresponding score was given (*Dong et al., 2020*). The total score represented the final score for the nozzle at the angle.

## Measurement of SA and MSD on the nasal septum

The surface of the nasal septum is flat and the SA can be easily measured. SA represents the region of distribution of the nasal spray on nasal septum. MSD was defined as the distance from the nasal tip to the farthest point of staining in the sagittal plane on nasal septum.

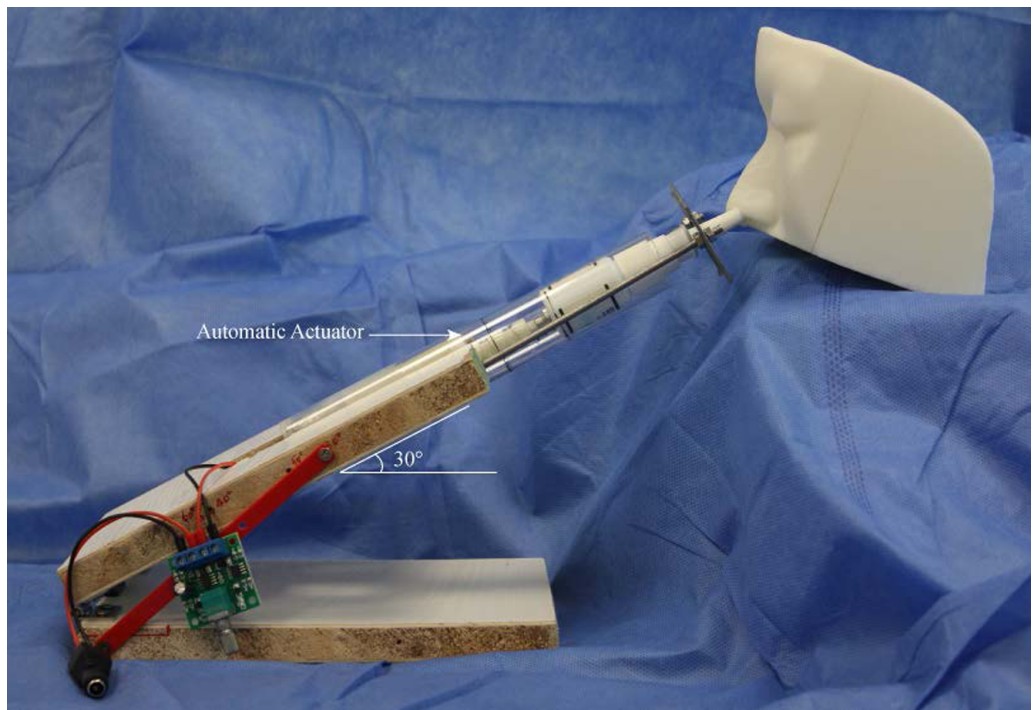

**Figure 3  Experimental apparatus for nasal drug delivery efficiency studies.** The testing spray device is placed into the automatic actuation station. The spraying angle can be adjusted at the desired administration angle.

After each test, the transparent paper was dried and scanned, and the SA was recorded and calculated automatically. The MSD on nasal septum was measured.

## Statistical analysis

Results were expressed as mean ± standard deviation (SD) or median with interquartile ranges. Statistical analyses were conducted using GraphPad Prism 8.0 (GraphPad Software, San Diego, CA, USA) and SPSS (version 22; IBM Corp., Armonk, NY, USA). Statistical significance was set at $P < 0.05$. ANOVA was used to determine the statistical significance of SA and MSD on the nasal septum. Scoring of the spray distribution on the lateral nasal wall was analyzed using the Kruskal–Wallis test. The $P$ values were adjusted using Bonferroni correction.

## RESULTS

### Characteristics of the different nozzles

After examination of spray patterns (Fig. 4), there were significant differences among the three commercially available nasal spray devices. The results are quantified in Table 1, and indicate that the SA of the three nozzles is significantly different. Nozzle C had the minimum spray area compared to nozzle A and B. Images of an emitted plume from the three nozzles are shown in Fig. 4. Plume angles derived from these photographs are shown Table 1. Comparison of plume angles indicated statistical differences among the three

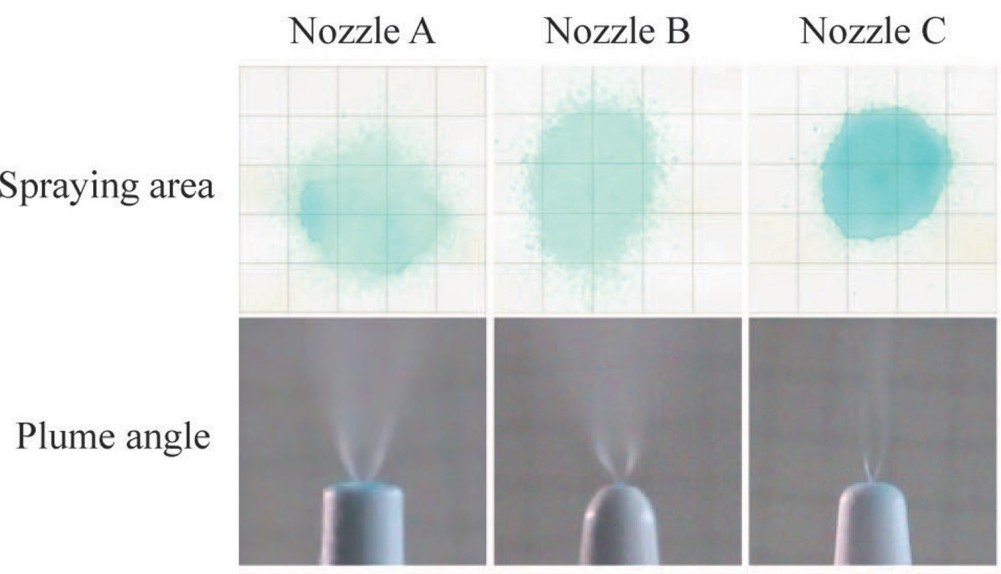

**Figure 4 Spray parameters measurement.** Spraying area images of the nasal spray devices at a perpendicular distance of four cm from the recording paper and plume angle images of different nozzles.

**Table 1 Comparison of characteristics of the different nozzles.**

| Parameters | Nozzle A | Nozzle B | Nozzle C |
|---|---|---|---|
| Volume of each spray ($\mu$L) | 67 | 50 | 80 |
| Distance of compression (mm) | 6 | 5 | 7 |
| Spraying area outside the nose (mm²) | 838.02 ± 43.54 | 924.80 ± 128.37 | 588.10 ± 19.88[*],[**] |
| Plume angle (°) | 49.47 ± 1.38 | 62.67 ± 2.38[*] | 37.47 ± 1.75[*],[**] |

**Notes.**

Data are expressed as means ± standard deviation, $n = 6$.

[*]Compared to Nozzle A, $P < 0.001$.

[**]Compared to Nozzle B, $P < 0.001$.

nozzles. Nozzle B had the maximum plume angle, and nozzle C had the minimum plume angle.

## SA on nasal septum

To determine the drug delivery efficiency at different angles of administration, we measured SA on nasal septum using spraying angles of 30°, 40°, and 50° with the three nozzles (Fig. 5A). As shown in Fig. 5B, with the increase in spraying angle of the three nozzles, the SA on nasal septum gradually increased. When the angle of administration was 50°, each nozzle reached the maximal SA. For nozzles A and C, the SAs at 40° and 50° increased significantly compared with those at 30° ($P < 0.05$). Although there was no statistically significant difference in SA among the different administration angles from nozzle B ($P > 0.05$), the average SA increased by 16.3% when the angle of administration increased from 30° to 50°. These results indicated that the larger the angle of administration, the higher the drug delivery efficiency within a certain angular range.

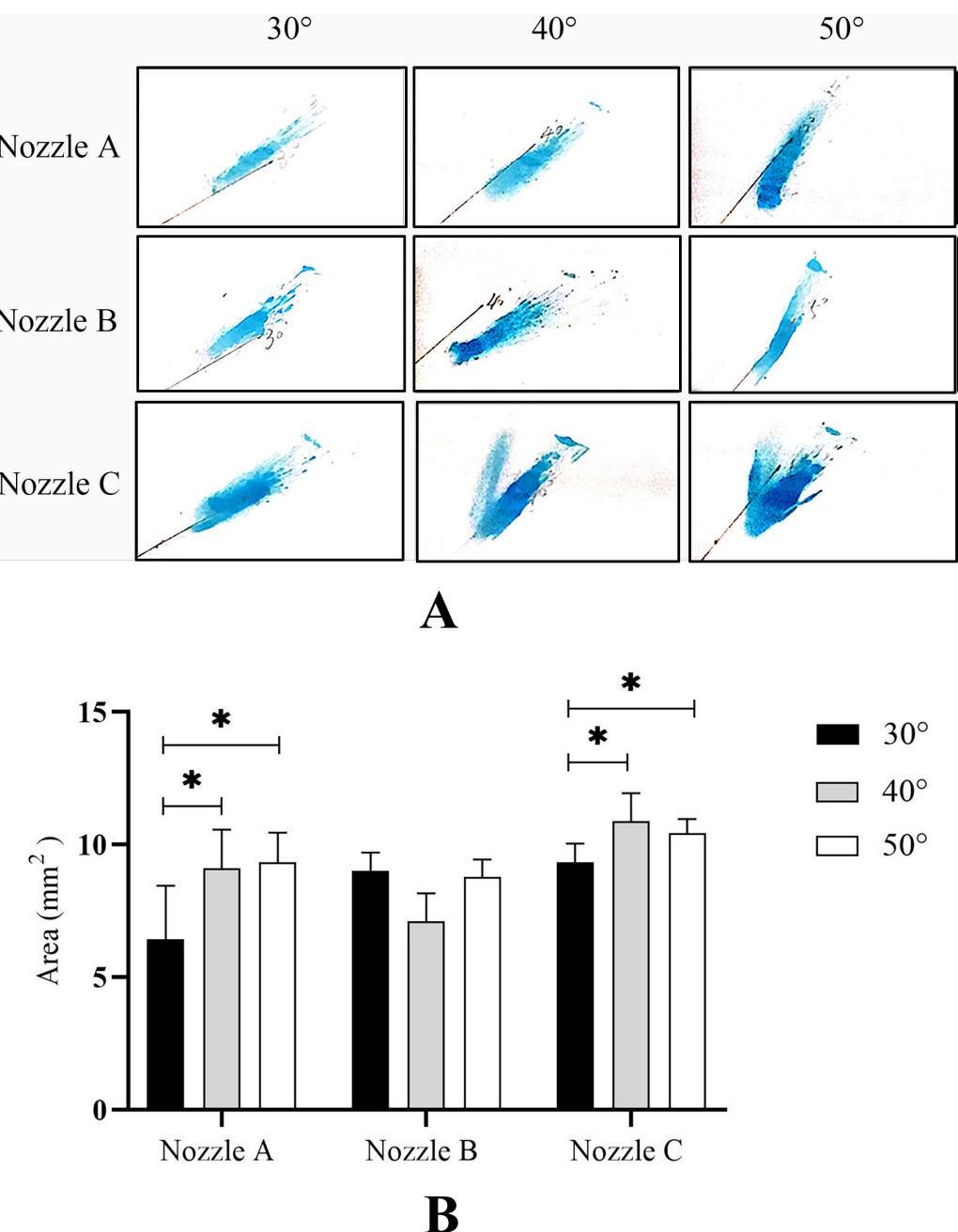

**Figure 5  Spraying area (SA) on the nasal septum.** (A) A representative image of nine independent experiments is shown.(B) Data are expressed as means ± standard deviation ($n = 9$). *$P < 0.05$.

## MSD on nasal septum

To understand the delivery efficiency of different spray devices on nasal septum, the MSD was measured at different administration angles. In this nasal cast model, the distance from the nasal tip to the olfactory cleft was 7.1 cm, to the sphenoethmoidal recess 7.5 cm, and to the posterior wall of the nasopharynx 8.0 cm. After nine times testing at each angle,

**Table 2  Maximal spraying distance on nasal septum (mean ± SD, cm).**

|  | Angle of administration | | | P |
|---|---|---|---|---|
|  | 30° | 40° | 50° |  |
|  | ($n = 9$) | ($n = 9$) | ($n = 9$) |  |
| Nozzle A | 7.00 ± 1.21 | 7.06 ± 0.68 | 6.91 ± 0.40 | 0.934 |
| Nozzle B | 7.12 ± 0.41 | 6.89 ± 0.86 | 6.97 ± 0.37 | 0.704 |
| Nozzle C | 7.56 ± 0.68 | 7.44 ± 0.17 | 7.10 ± 0.00 | 0.095 |
| P | 0.361 | 0.200 | 0.445 |  |

**Notes.**
Data are expressed as means ± standard deviation.

there was no statistically significant difference in MSD among the three nozzles at different angles (Table 2).

### Spraying distribution on lateral nasal wall

Because of the complexity of anatomy on the lateral nasal wall, SA is difficult to measure accurately. We evaluated the delivery efficiency on the lateral nasal wall using the stain scoring (Fig. 6A). In descending order, the total scores for each nozzle using the different spraying angles were as follows: nozzle A, 40° > 30° > 50°; nozzle B, 30° > 40° > 50°; and nozzle C, 30° > 40° > 50° (Table 3). However, the differences in total scores at different angles when using nozzle C were statistically significant ($P < 0.05$). Interestingly, the total scores for different nozzles with the same angle were significantly different, and nozzle C had the highest total scores. It is worth noting that none of the three nozzles could effectively deliver the spraying solution into the middle meatus at any angle (Fig. 6B).

## DISCUSSION

Due to the complexity of nasal structures, drug delivery efficiency in nasal cavity is difficult to assess. Although *in vivo* assessment of humans is preferred, *in vitro* evaluation using nasal cast models has been proposed as a low cost and speed means of assessment of bioequivalence of delivery between products (*Djupesland, Messina & Mahmoud, 2020*). In this study, the drug delivery efficiency of three nasal spray devices was evaluated in three dimensions using a 3D printed nasal cast model, namely SA, MSD, and scoring of spray distribution.

Nasal cast models are helpful approximations for early product development, but inadequate for accurate simulation of the human nasal cavity *in vivo* (*Djupesland, Messina & Mahmoud, 2020*). Currently, there are two human nasal silicone casts are used for *in vitro* evaluation of nasal drug delivery, the Koken cast (Koken Co., Tokyo, Japan) and the Optinose cast (OptiNose AS, Oslo, Norway) (*Castile et al., 2013*; *Palmer et al., 2018*). The Koken cast is commercially available and is based on a cadaver. It is primarily an educational tool and described as 'anatomically correct' (*Djupesland, Messina & Mahmoud, 2020*). The Optinose cast was developed with 3D computer reconstruction and the intranasal surface was rendered from high resolution MRI of a healthy male. These casts were dimensionally compared and assessed for deposition assessment suitability

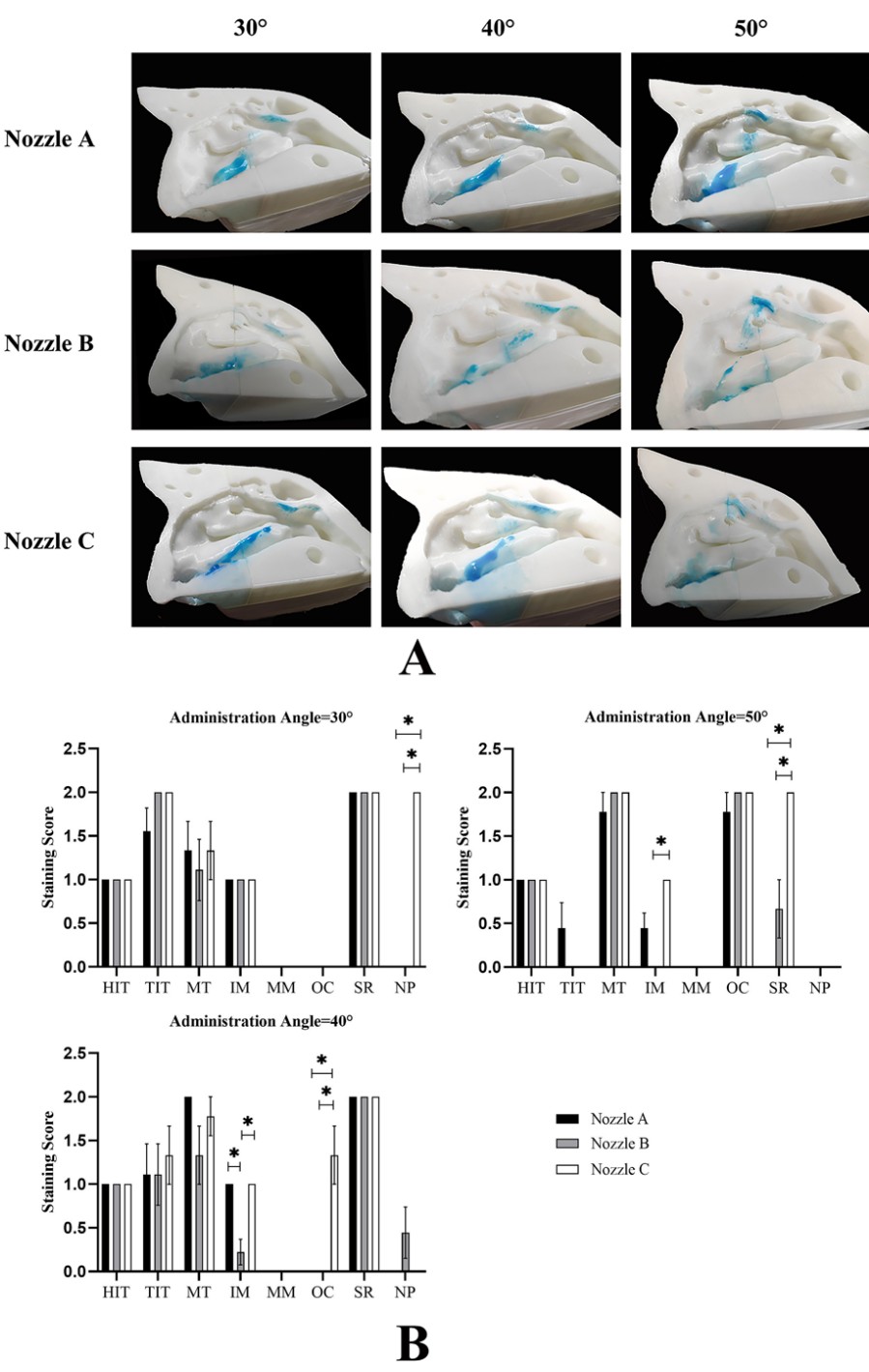

**Figure 6  Distribution of spray on the lateral nasal wall.** (A) A representative image of nine independent experiments is shown. (B) Data are presented as medians with interquartile ranges. ($n = 9$). *$P < 0.05$.

**Table 3  Staining score on the lateral nasal wall.**

| Group | Angle of administration | | | u | P |
|-------|------------------|------------------|------------------|---|---|
|       | 30° ($n=9$) | 40° ($n=9$) | 50° ($n=9$) |   |   |
| Nozzle A | 8 [6,8] | 8 [6,8] | 6 [5,6] | 4.497 | 0.106 |
| Nozzle B | 8 [6,8] | 6 [5,7] | 5 [5,7] | 5.929 | 0.052 |
| Nozzle C | 10 [8,10] | 8 [8,10] | 8 [8,8] | 7.644 | 0.022* |
| u | 13.399 | 8.439 | 16.887 | | |
| P | 0.001* | 0.015* | 0.001* | | |

**Notes.**
Data are presented as medians with interquartile ranges [lower, upper quartile].
$u$ is reflective of a standardized test statistic for the Kruskal–Wallis test.
*$P < 0.05$, statistically significant difference between groups.

(*Djupesland, Messina & Mahmoud, 2020*). The 3D model in our study was also constructed based on CT data and could well simulate the structure of the nasal cavity and sinuses. It consists of four components that can be split for easy observation. However, the printing material is polylactic acid, instead of silicone. The hardness of the model is higher than that of the above two molds, which could compromise the real simulation effect.

Previous studies focused on the effects of the number of nozzles, the characteristics of pharmaceutical formulations, and the mode of administration on drug delivery efficiency (*D'Angelo et al., 2023*; *Djupesland, Messina & Mahmoud, 2020*; *Dong et al., 2020*; *Kapadia, Grullo & Tarabichi, 2019*; *Sosnowski et al., 2020*; *Tong et al., 2016*). In this study, we try to explain the effects of nozzle design on drug delivery efficiency. By measuring the SA on nasal septum, the same trend was observed among the different nozzles; that is, with an increase in spraying angle, SA gradually increased within a certain angular range (30°–50°). However, we did not reach the same conclusion regarding the evaluation on the lateral nasal wall. There was no statistically significant difference in the total scores for the nozzles under different spraying angles, apart from nozzle C. We also compared the MSD on the nasal septum, and there was no statistically significant difference among the three nozzles. These nasal spray devices can easily deliver drugs to most regions of the nasal cavity, such as the turbinate, nasal septum, olfactory fissure, and nasopharynx, but not the middle meatus. Our findings were consistent with those of previous studies that both the squeeze bottle and soft-mist nasal pump yielded notably low doses to the ostiomeatal complex with high variability, and no dose from these two devices was detected within the maxillary sinuses (*Seifelnasr et al., 2023b*). It is feasible to deliver clinically significant doses of nasal sprays to the target region by leveraging an optimized combination of delivery variables (*Seifelnasr, Si & Xi, 2023a*).

The plume angle of each nozzle determines the SA of the nasal spray device. Plume angle is proportional to the SA outside the nose (Table 1). It has been reported that, with the same plume angle, the nasal deposition efficiency with different spraying angles was 50° <40° <30°[3]. This conclusion is contradictory to our findings. In earlier study, the nasal model based on MRI data was used and the turbinate deposition efficiency was the primary evaluation parameter. Two points should be noted in this regard. The degree of

nasal septum deviation has a significant influence on the distribution of drugs in the nasal cavity (*Frank et al., 2012a*; *Frank et al., 2012b*). While, in this study, we used transparent paper to replace the nasal septum to minimize its interference and facilitate observation. In addition, the observation parameters we chose were different from those used in the previous study. Deposition efficiency measuring the percentage volume of drug entering the nasal cavity in the total volume of drug sprayed each time ignores the distribution of drugs in the nasal cavity. Drug overlap can lead to a reduction in drug distribution. However, SA combined the scoring system can objectively reflect the distribution of drugs in the nasal cavity. These factors could explain why our conclusion was inconsistent with that of the previous study.

To the best of our knowledge, SA on nasal septum has never been used to evaluate drug delivery efficiency. *Kundoor & Dalby (2011)* quantitatively calculated the dye projection area in the nasal cavity using Adobe Photoshop (Adobe, San Jose, CA, USA) and found that lower angles of administration had a lower deposition area compared to higher angles of administration. Interestingly, their conclusions were consistent with our findings. However, their assessment method ignored the overlap between the lateral nasal wall and nasal septum, resulting in a smaller calculated area than the actual area. MSD was measured to fully evaluate the longitudinal distribution of the drugs on the nasal septum. Using this model, these three nozzles could deliver drugs to the deepest part of the nasal cavity at any angle.

The lateral wall of the nasal cavity has complex three-dimensional structures, and the SA on the lateral nasal wall is difficult to measure accurately. To overcome this difficulty, we introduced a scoring system to evaluate the delivery efficiency on the lateral nasal wall depending on how easily the pigment spreads in the nasal cavity. As shown in Table 3, the highest scoring for each nozzle angle came from the lowest administration angle, and Nozzle C had the highest score at any of the angles of administration among the three nozzles, which has the smallest plume angle. Interestingly, after examining the pigment distribution on the lateral nasal wall, we found that the scoring for the middle nasal meatus of any nozzle at any angle was zero (Fig. 6B). Using iopamidol-labeled nasal spray in a 3D model, *Sartoretti et al. (2019)* reported that areas with high radiation doses were mainly concentrated in the olfactory area and turbinate, and only a small amount of radiation was detected in the middle meatus. This conclusion needs to be further verified in future clinical trials. Many factors affect the efficiency of drug delivery to the middle meatus, including the turbinate morphology, nozzle design, angle of administration, and physicochemical properties of nasal spray products. *Kapadia, Grullo & Tarabichi (2019)* compared the delivery efficiency of long and short nozzles on 14 cadavers and found that the long nozzle was more effective than the short nozzle in delivering drugs to the sinuses. *Pennington et al. (2008)* found that spray area and droplet size were related to drugs viscosity. The particle size of 10 μm were the most suitable as the majority of agents can be delivered to the targeted area (*Tong et al., 2016*). Droplet velocity is another critical parameter in the deposition in the nasal cavity and droplets with velocities <1 m/s can produce a more uniform coverage (*D'Angelo et al., 2023*). *Pourmehran et al. (2022)* demonstrated that increasing the flow swirl at the inlet and decreasing the nozzle diameter improves the total particle deposition.

There were limitations in this study. First, we did not assess the interfering effect of nasal airflow and physicochemical properties of nasal spray products on nasal drug distribution. In clinical practice, nasal airflow plays an important role on drug delivery efficiency. Second, this 3D printed model is an idealized model, which is derived from a CT scan of a healthy volunteer. In fact, majority of the population have associated nasal abnormalities, which significantly affect the nasal drug distribution. Finally, the relative position of the nasal cavity and the nozzle also has a certain impact on the drug distribution in nasal cavity. For example, clinicians usually direct the patient to point the nozzle toward the lateral nasal wall. Further studies are needed to identify the drug delivery efficiency in practical clinic application.

## CONCLUSIONS

This study provided quantitative measurements of the effects of three nasal spray devices on a 3D printed model from multiple dimensions. Due to limitation in the internal nasal valve, the ideal angle of administration is 50°. Apart from the middle meatus, the current nasal spray devices can easily deliver drugs to turbinate, nasal septum, olfactory fissure, and nasopharynx. Nozzle C with smaller plume angle can deliver more medication to the lateral wall of nasal cavity. Therefore, the efficiency of drug delivery is closely related to the angle of administration and nozzle design. Recently, we designed a nasal continuous drug delivery device, which is supposed to deliver the drug precisely to the designated site of the nasal cavity. However, its delivery efficiency needs to be further verified. In addition, highly simulated nasal cast models facilitate accurate *in vitro* testing of nasal spray devices. These results can provide theoretical basis for nozzle selection and the design of nasal drug delivery device in the future.

## ACKNOWLEDGEMENTS

We thank Mr. Zhonghan Wang for setting up the experimental facility.

### Funding

This work was supported by the Shanghai Municipal Health Commission Clinical Research Special Project (No. 202150005) and the Shanghai East Hospital Reserve Personnel Training Program (No. 2019YHRCJH02). The funders had no role in study design, data collection and analysis, decision to publish, or preparation of the manuscript.

### Grant Disclosures

The following grant information was disclosed by the authors:
Shanghai Municipal Health Commission Clinical Research Special Project: 202150005.
Shanghai East Hospital Reserve Personnel Training Program: 2019YHRCJH02.

### Competing Interests

The authors declare there are no competing interests.

## Author Contributions

- Shengjian Fang conceived and designed the experiments, performed the experiments, prepared figures and/or tables, authored or reviewed drafts of the article, and approved the final draft.
- Xiaoqing Rui performed the experiments, authored or reviewed drafts of the article, and approved the final draft.
- Yu Zhang analyzed the data, prepared figures and/or tables, and approved the final draft.
- Zhangwei Yang analyzed the data, prepared figures and/or tables, and approved the final draft.
- Weihua Wang conceived and designed the experiments, performed the experiments, prepared figures and/or tables, authored or reviewed drafts of the article, and approved the final draft.

## Data Availability

The raw measurements are available in the Supplementary Files.

## Supplemental Information

Supplemental information for this article can be found online at http://dx.doi.org/10.7717/peerj.17227#supplemental-information.

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
