# Peer review of "Comparative study of nasal cavity drug delivery efficiency with different nozzles in a 3D printed model"

_PeerJ, doi:10.7717/peerj.17227_

## Round 0.1 · original submission · Major Revisions

An interesting study. However, more attention needs to be paid to the experimental design. This must be improved.

Reviewer 1 ·

Basic reporting

This manuscript describes a comparative study of nasal cavity drug delivery efficiency with different nozzles in a 3D-printed model. The overall article is interesting, and the paper is well-written and clear; however, the manuscript needs some significant revisions to be considered for publication. Below are some points that might help improve the quality of this work and presentation.

1) The authors must improve the literature review of the manuscript. There are no works from 2020 which are cited, and only one work from 2021 and 2022 are cited. This points to an inadequate literature survey; therefore, the authors are suggested to perform an extensive review and update the manuscript accordingly.

2) The introduction section does not provide adequate background about how this work advances the field or contributed, addressing point number 1, and also overhauling and extensively expanding this section would help the authors achieve that. In its present state, the introduction section is severely lacking and limited.

3) The authors are highly advised to create a graphical abstract of this work and include it with the manuscript; this would significantly improve the readability of the work and also make it further interesting to the broader audience.

Experimental design

4) Unfortunately, the materials and methods section is not detailed enough for readers to replicate the authors work. Each and every material and method used in this manuscript must be described in detail, and their origin or source must be mentioned. The model, product number, or any other manufacturer identifiers should be mentioned for the devices and the chemicals used.

5) The procedure for creating the 3D-printed model from the CT data must be described in detail. This is missing. Also, the 3D printer used (UP BOX has different models of printers), the filament type that is used, the manufacturer of the filament, etc. must be described in detail.

6) What are the questions and hypotheses that the authors have and are trying to address? Even though this is an application-focused paper, it must have these questions and hypotheses well defined and be explicitly stated; again, addressing the many points raised might help improve this.

Validity of the findings

7) The conclusion section is incomplete. A mere 4 line conclusion section that does not describe or discuss the implications of the study or future outlook would not be suitable for a manuscript that can be considered for publication. Please make the conclusion section detailed, tie it with your research, and discuss the implications of your findings and future outlook based on your study.

8) Throughout the manuscript, in various sections, it feels as if the authors have not provided all possible data or performed the necessary experiments. The authors also mention in the limitations that they have not looked at the physicochemical properties of the nasal spray products, which in my opinion, could have been easily performed and could add significantly to the value of the work. The authors should also perform control studies and add those data to future submissions.

Additional comments

Please describe what you mean by "technical help" in line 247. This would help readers understand the exact contribution of the acknowledged person.

Make sure all your references are uniformly formatted and are consistent throughout.

·

Basic reporting

The author used a 3D-printed nasal model to test the drug delivery efficiency of three different nozzles. The language and structure of the paper are good.

Experimental design

The experimental design need to be improved:
1) Only the nozzle direction paralleling to the sagittal plane was tested. However, the nasal spray should be used towards the lateral wall of the nasal cavity of the patient in the real world.
2) The hardness of the printing material should be mentioned. Is it comparable to the soft tissue of the nose?

Validity of the findings

Please explain more about the soring system. Why the different scores were used in different parts of the nasal cavity?

Additional comments

I do not think the CT image is necessary for publication.

---

## Round 0.2 · Minor Revisions

This article needs further revision. In addition, statisticians are required to detect which statistical analysis is performed to the technical standard (e.g. Table 1-3, sample size, use of appropriate statistical measures, correction for multiple testing, effect size, degrees of freedom, etc.)

Reviewer 1 ·

Basic reporting

Refer "4. Additional comments"

Experimental design

Refer "4. Additional comments"

Validity of the findings

Refer "4. Additional comments"

Additional comments

The authors have addressed most of my comments, however, the response is not adequate. For some of the comments, the authors have responded to my comments but they have not addressed the deficiency in the paper. Comment 6 is an example.

The conclusion section is still lacking.

The literature survey is still not adequate. For example, only one work from 2023 is cited.

Figure 1 text needs grammatical corrections. Also, label the components of Figure 1.

Please go through the comments from the previous round, and address them within your manuscript carefully.

Make sure every section is detailed enough for others to reproduce your work.

·

Basic reporting

no comment

Experimental design

no comment

Validity of the findings

no comment

---

## Round 0.3 · Minor Revisions

Please expand the literature review as requested by the reviewer.

Reviewer 1 ·

Basic reporting

.

Experimental design

.

Validity of the findings

.

Additional comments

One recommendation that I still have is that the literature survey (Introduction section) still needs to improve.

---

## Round 0.4 · Minor Revisions

To further improve the quality of the article, additional editorial comments from the Section Editor are conveyed below.

> Where is CT scans with embedded scale in Fig 1? Looking closely, it looks like pixels have been altered. There is also no scale in any other image.
Figure and table legends could be better. And in Fig 6 B, no statistical differences.

---

## Round 0.5 · accepted · Accept

The authors have addressed all of the comments.